# Process Mineralogy of Micro/Nano Silicon-Carbon Ore Obtained from Jiangxi, China

Shaowen Tang [1] , Huan Shuai [1] , Runtao Zhao [1], Gaoxiang Du [1,2,*], Xianguang Wang [3] and Jiao Wang [4]

1 School of Materials Science and Technology, China University of Geosciences, Beijing 100083, China; 2103200011@cugb.edu.cn (S.T.); shuaihuan@email.cugb.edu.cn (H.S.); 2003200005@cugb.edu.cn (R.Z.)
2 Beijing Yiyi Star Technology Co., Ltd., Beijing 100089, China
3 Jiangxi Mineral Resources Guarantee Service Center, Nanchang 330025, China; 13907090885@163.com
4 School of Basic Education, Beijing Polytechnic College, Beijing 100042, China; wj@bgy.edu.cn
* Correspondence: dgx@cugb.edu.cn

**Abstract:** This study details the investigation of a black powder potential ore that was first obtained from Jiangxi, China. Its species, composition, and morphology are unknown. Preliminary tests revealed that the silica ($SiO_2$) content of this ore is >70%. To test this ore more comprehensively, its mineralogical parameters (such as mineral composition, ore particle size, and mineral morphology) are investigated using X-ray diffraction (XRD), inductively coupled plasma mass spectrometry (ICP-MS), differential scanning calorimetry–thermogravimetry (DSC-TG), Fourier-transform infrared spectrometry (FTIR), scanning electron microscopy (SEM), laser particle size analysis, and elemental analysis (EA). Based on these analyses, it is determined that it is micro/nano silicon-carbon ore, and its genesis and species are discussed herein. The gangue minerals, such as α-quartz, kaolinite, pyrite, and muscovite, are finely disseminated and encapsulated by fixed carbon. The ore has an uneven morphology, with many holes and depressions. Moreover, nano-sized needle-like quartz and quartz wrapped by carbon are found on the surface of the ore. According to our results, this ore may have been formed by the long-term accumulation and consolidation of phytoliths. These results provide a technical reference for the development and utilization of the identified micro/nano silicon-carbon ore.

**Keywords:** process mineralogy; micro/nano silicon-carbon ore; silica; characterization; phytolite

## 1. Introduction

Process mineralogy is an integrated discipline that combines quantitative (and qualitative) mineralogical techniques with metallurgical test work [1]. It plays a fundamental role in the planning, optimization, and monitoring stages for different types of ores [2]. The exploitation of new mineral resources and the reuse of tailings both depend on research into the various process mineralogy parameters of different ores [3–6]. However, different ores can have differing compositions, morphologies, particle size distributions, mineral liberation degrees, and textural relationships. Owing to these features, it is necessary to use different complementary analytical techniques, such as X-ray diffraction (XRD), X-ray fluorescence spectrometry (XRF), scanning electron microscopy (SEM), Fourier-transform infrared (FTIR) spectrometry, optical microscopy, and inductively coupled plasma mass spectrometry (ICP-MS) [7–11]. To date, various process mineralogy research methods have been used to analyze a variety of ores, and a range of different beneficiation methods have been designed [12–15].

This study focuses on a black powdery potential ore that was originally obtained from Fengcheng, Jiangxi, China. This ore was found in a mining area named Shilukeng. Clay minerals, such as montmorillonite, kaolin, illite, and pyrite, have also been mined in Shilukeng. The ore is found in an area of 1.83 km², and the average thickness is 30.53 m. This study details how process mineralogy methods were applied to characterize

the species, characteristics, and mineral composition of this ore, which were previously unknown. Herein, ICP-MS, elemental analysis (EA), differential scanning calorimetry (DSC), FTIR, and XRD were used to determine the mineral composition of the ore. SEM and laser particle size analysis were used to explore its morphology and particle size. The genesis and application potential of the ore were then discussed based on these test results. We analyzed the genesis and application field of the unknown ore by studying its mineral composition and morphology. Moreover, we aimed to provide a theoretical basis for the investigation, purification, and utilization of the ore in question.

## 2. Material and Methods

### 2.1. Ore Sample and Reagents

The ore sample studied herein was obtained from a mining hole in Fengcheng, Jiangxi, China (Figure 1). It comprised a stable black powder that could not be ignited, and some parts of the sample were less dense than water. For testing and research, a 5 kg ore sample was dried in an oven at 60 °C, then pounded and mixed to obtain a uniform composition for ensuring accurate results. Sodium hydroxide (NaOH, AR; Shanghai Aladdin Bio-Chem Technology Co., Ltd.) was used to remove quartz and silicate from the ore. A sodium hypochlorite (NaClO, CP, 5%; Beijing Sinopharm Chemical Reagent Co., Ltd., Beijing, China) solution was used to remove carbon.

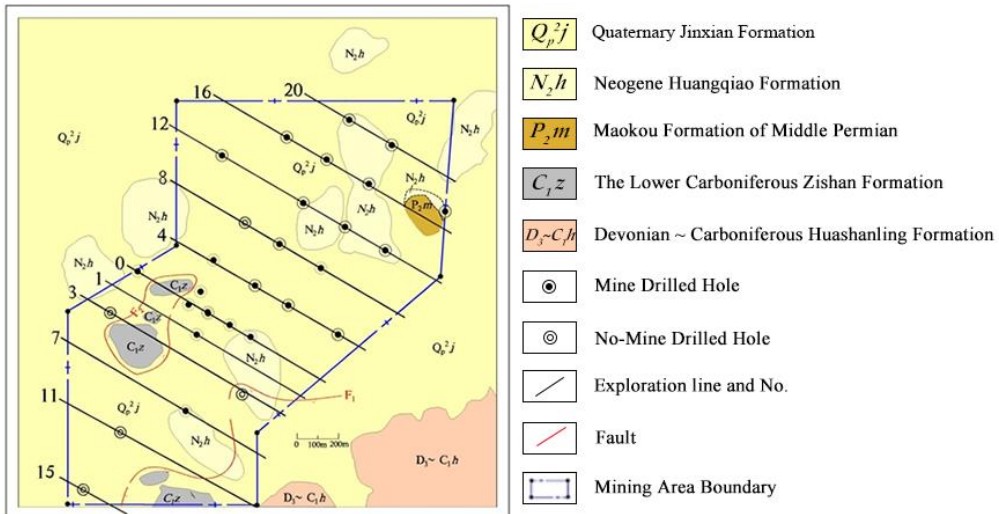

**Figure 1.** Geology map of Shilukeng deposit.

### 2.2. Sample Preparation for Carbon/Quartz Removal

In addition to the original sample, two chemically treated samples of the ore were examined to ensure comprehensive investigation of its microstructure. In these treated samples, carbon and silicon were removed.

High-temperature calcination was not used for carbon removal to ensure that the quartz structure is not disturbed. Carbon removal was carried out via oxidation with NaClO at 60 °C. First, the ore sample was added to 200 mL of a NaClO solution (5%). After magnetic stirring at 60 °C for 4 h, the sample was washed and filtered. The above steps were repeated three times, and thereafter, the carbon-free sample was obtained in the form of a white powder.

A hydrothermal method was used to remove silicon ($SiO_2$) from the sample, based on the following chemical reaction between quartz and NaOH: $nSiO_2 + 2NaOH \rightarrow Na_2O \cdot nSiO_2 + H_2O$. This method enabled the safe removal of quartz, without impacting the carbon structure. The ore sample (5 g) was mixed evenly in 50 mL of a NaOH solution (4 mol·L$^{-1}$). The resulting turbid liquid was poured into a reaction kettle and reacted in an oven at 200 °C for 2 h. After 2 h, the oven was turned off, and when the temperature decreased to

approximately 80 °C, the reaction kettle was removed. The product was then filtered and dried to obtain a black powder containing no quartz.

### 2.3. Multi-Element Analysis

ICP-MS (Agilent 7500A, Agilent Corporation, Santa Clara, CA, USA) and EA (Vario MACRO Cube, Elementar, Langenselbold, Germany) were used to determine the elemental composition of the ore sample. Soil (GSS-2) reference material (Hunxin Measurement and Testing Co., Ltd., Blue Mountains, NSW, Australia) was used to calibrate the ICP-MS equipment. The ore sample was acid-digested with hydrochloric and hydrofluoric acids for ICP-MS analysis. Quantitative analysis of chemical elements was then conducted with RF power = 1260 W at a carrier gas flow rate of 1.23 L·min$^{-1}$. The detection limit was 0.010 μg·g$^{-1}$. The sample (20 mg) for EA was prepared by drying the ore sample in an oven at 80 °C.

### 2.4. Differential Scanning Calorimetry–Thermogravimetry (DSC-TG)

The ore sample (10 mg) was placed in a Pt crucible and heated from 30 to 1000 °C at a heating rate of 10 °C·min$^{-1}$ using DSC (NETZSCH STA 449F3, NETZSCH, Selb, Germany).

### 2.5. FTIR Spectrometry Measurements

The ore sample (2 mg) was mixed with potassium bromide (KBr, 100 mg) powder and compressed into tablets. An FTIR spectrometer (TENSOR27, Bruker, Karlsruhe, Germany) was used to qualitatively analyze the organic matter. The measurement was conducted for 16 s in the 400–4000 cm$^{-1}$ range with 4 cm$^{-1}$ resolution.

### 2.6. XRD Detection

The sample for XRD measurements was prepared using the same steps used for FTIR sample preparation. The ore sample was scanned using an X-ray powder diffractometer (D8 Advance, Bruker, Germany) with a Cu target ceramic light tube to identify mineral phases. The scanning range (2θ) was 10–70°, and the scanning speed was 2°·min$^{-1}$.

### 2.7. SEM Characterization

Three samples (original ore, carbon-free, and silicon-free) were analyzed using SEM to observe their morphologies. Each sample was dispersed in alcohol and then dropped on an aluminum foil. After the alcohol had reacted, the aluminum foil was glued to the sample table with a carbon-conductive adhesive. Then, Au was sprayed onto the surface of each sample to enhance its electrical conductivity. SEM (SIGMA 300, ZEISS, Jena, Germany) and EDS (INCA X-ACT, Oxford Instruments, Abingdon-on-Thames, UK) were then used to observe the samples (acceleration voltage = 15 kV). Carbon black (N330) reference material was used to calibrate the EDS.

### 2.8. Particle Size Testing

A laser particle size tester (Model Bettersize2000, Dandong Bettersize Instruments, Dandong, China) was used to conduct particle size analysis. The latex particle reference material (PS-DVB, 5.4 μm) was used to calibrate the tester. The three types of samples (original ore, carbon-free, and silicon-free) were analyzed. The dispersing medium was water, and the dispersant was sodium hexametaphosphate (6 wt%).

## 3. Results

### 3.1. Mineral Composition

The results of multi-element analysis of the ore sample are shown in Table 1. The XRD pattern of the ore sample is shown in Figure 2. Table 1 shows that the main component of the ore was $SiO_2$ (81.28%), which originated from quartz. Carbon accounted for 10.77% of the ore sample. The mass fraction of $Al_2O_3$ in the ore was 4.56%, which mainly originated from kaolinite. Iron and sulfur were mainly present in the form of pyrite (2.01%). The

ore also contained small amounts of Mg, Ti, Ca, K, and Na, derived from minerals such as muscovite.

**Table 1.** Multi-element analysis of the ore sample by ICP-MS and EA.

| Element | $SiO_2$ | $Al_2O_3$ | MgO | $TiO_2$ | CaO |
|---|---|---|---|---|---|
| Content (wt%) | 81.28 | 4.56 | 0.15 | 0.32 | 0.19 |
| **Element** | $K_2O$ | $FeS_2$ | $Na_2O$ | C | |
| Content (wt%) | 0.44 | 2.01 | 0.28 | 10.77 | |

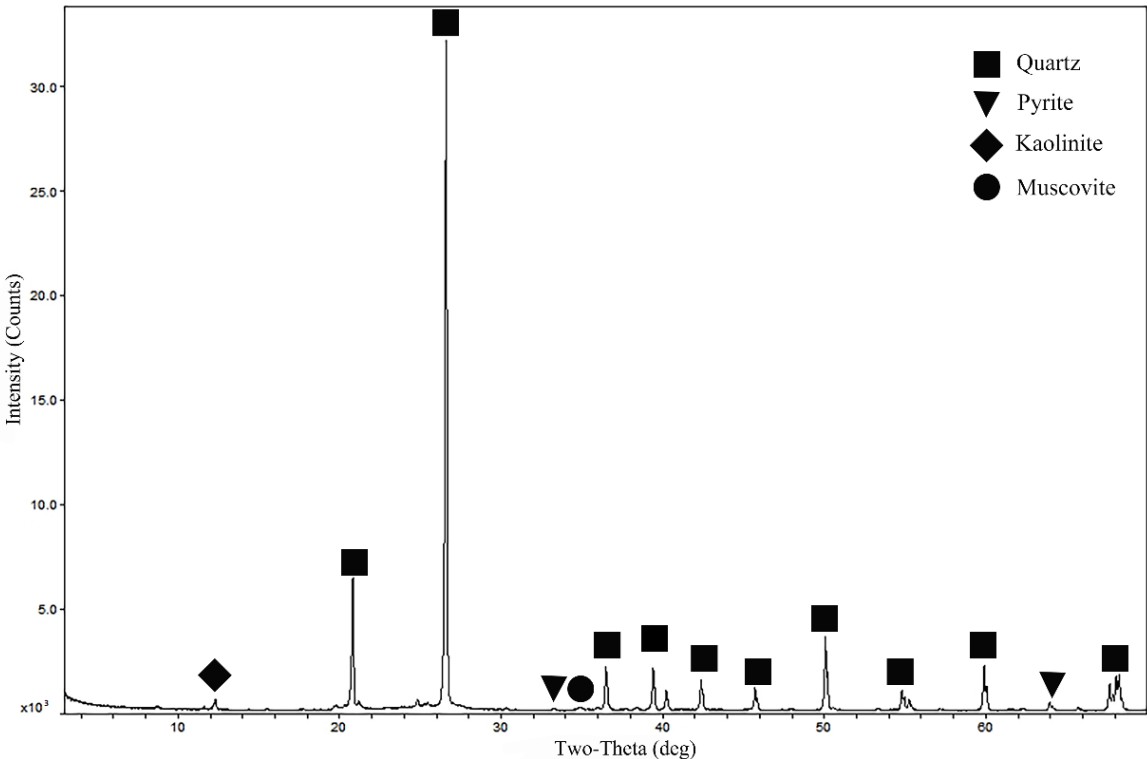

**Figure 2.** XRD spectra of the ore sample.

Figure 3a shows the DSC-TG curves of the ore sample. Across the entire reaction, the quality of the ore decreased by 10.79%. The first decomposition stage of the sample ranged from 25 to 200 °C. This stage comprised an endothermic reaction; it presented as a convex curve. Moisture and volatile compounds were released from the mineral sample during this stage [16]. The endothermic peak was located at 64.3 °C, and the chemical reaction heat was 0.656 mW·mg$^{-1}$. The second decomposition stage of the sample ranged from 420 to 720 °C. This stage comprised an exothermic reaction; it exhibited a sunken curve. This stage mainly comprised the oxidation of carbon or organic compounds in the mineral sample [17]. The exothermic peak at this stage was at 625.9 °C, which is when the oxidation of carbon reached its highest reaction rate. The chemical reaction heat was −7.531 mW·mg$^{-1}$.

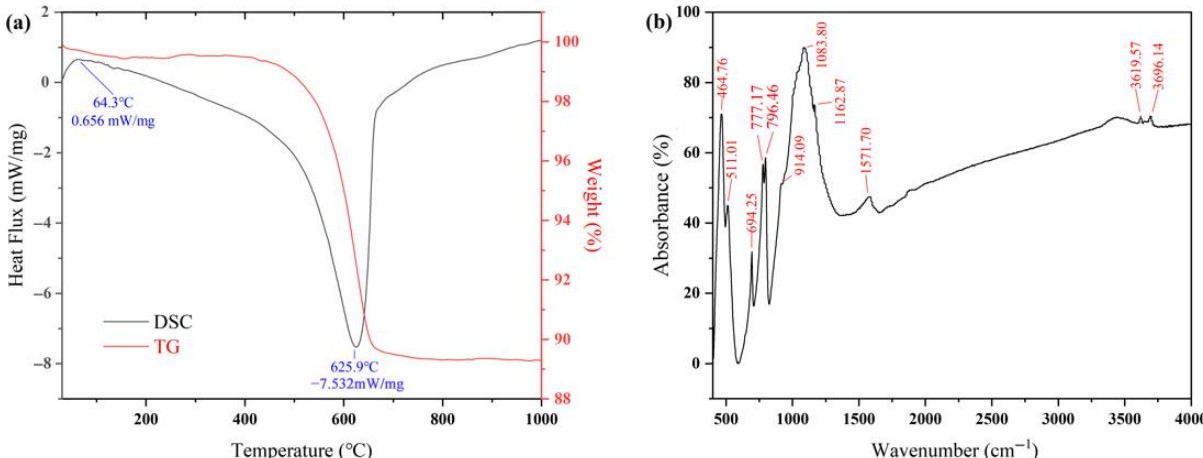

**Figure 3.** (**a**) DSC-TG curves and (**b**) FTIR spectra of the ore sample.

Figure 3b shows the FTIR pattern of the ore sample. The characteristic absorption peaks of hydroxyl (O-H) stretching vibration appeared in the range of 4000–3500 cm$^{-1}$ (Figure 3b), with wavenumbers of 3696.14 and 3619.57 cm$^{-1}$. Combined with DSC analysis, these characteristic absorption peaks were attributed to small amounts of water and hydroxyl in the mineral sample [18].

A single characteristic absorption peak was observed at 1571.70 cm$^{-1}$. This corresponded to the stretching vibration of a carbon-carbon single bond (C-C).

Eight characteristic absorption peaks were identified in the range of 400–1330 cm$^{-1}$. The double peaks at 1162.87 and 1083.80 cm$^{-1}$ were recognized as Si-O-Si antisymmetric stretching vibration absorption peaks, while the double peaks at 796.46 and 777.17 cm$^{-1}$ were classified as Si-O-Si symmetric stretching vibration absorption peaks. The double peaks at 511.04 and 464.76 cm$^{-1}$, however, were identified as the bending vibration absorption peaks of the Si-O bond [19]. The single characteristic absorption peak at 914.09 cm$^{-1}$ was classified as the Al-OH swing absorption peak, while the single absorption peak at 694.25 cm$^{-1}$ was assigned as the Al-OH vertical vibration absorption peak. Combined with the XRD results, these findings prove that the absorption peaks in this region mainly belonged to the absorption peaks of quartz and kaolinite [20]. The low modal abundances of pyrite, muscovite, and other minerals meant that they were not obviously reflected in the IR spectrum.

*3.2. Ore Morphology*

As shown in Figure 4a, SEM-EDS revealed many cracks and holes as well as circular pits and marks on the ore surface. These may have been caused by biodeterioration. As shown in Figure 4b, unevenly shaped micron-scale quartz pieces, nanoparticles, and nano-needles were found on the ore's surface. A small amount of nanoscale needle-like quartz crystals were also observed on the surface of the ore (Figure 4c). Figure 4d shows the elemental distribution of the ore surface over the marked area A, as determined by EDS analysis. The main elements in this ore were identified as C (42.51 wt%), O, and Si. Combined with the thermal analysis and XRD results, the mass percentage of carbon obtained by EDS was much higher than 10.77%. This means that carbon was mainly enriched on the surface of the mineral or in some special structures instead of being uniformly distributed throughout the mineral structure. A small amount of aluminum (4.64%) was also detected. This mainly came from aluminum silicates.

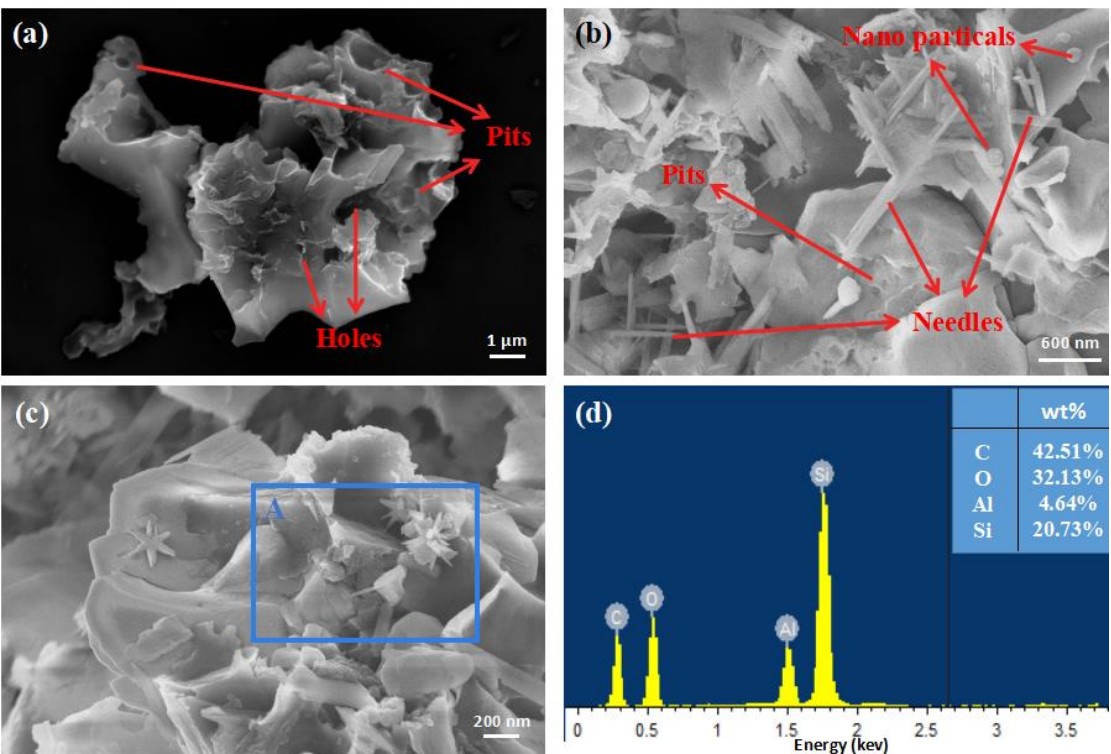

**Figure 4.** (**a**–**c**) SEM images of ore sample and (**d**) EDS elemental spectrum of area A.

The morphology of the carbon-free sample is shown in Figure 5a. After chemical treatment with NaClO, there were still some holes, cracks, and nano-sized quartz particles. However, no needle-like structures were found on the surface of the sample. To verify the carbon removal effect of NaClO, EDS analysis was performed on area B. As shown in Figure 5b, the main elements in the particle were O, Si, and Al, with O having the highest mass fraction (60.64%). Most of the carbon in the mineral was removed without damaging the basic structure of the quartz. This method was helpful for eliminating interference from fixed carbon and analyzing the original quartz morphology. It was also helpful for the purification of quartz at 60 °C.

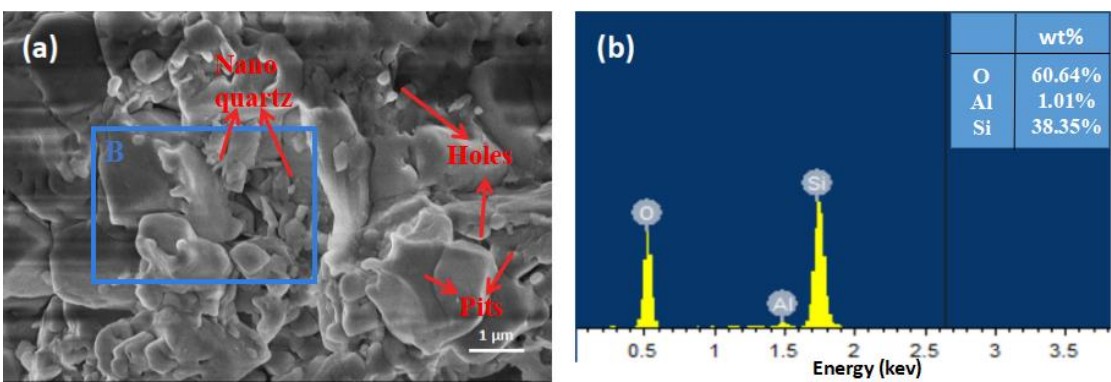

**Figure 5.** (**a**) SEM image of the carbon-free sample and (**b**) EDS pattern of area B.

The morphology of the quartz removal sample is shown in Figure 6a–c. Comparing Figure 6a with Figure 4b reveals that needle-like structures and holes remained in the sample after chemical treatment with NaOH. This means that the main element of this needle was carbon. Furthermore, Figure 6b,c show that the quartz in the sample was corroded by NaOH, resulting in large numbers of holes and depressions on the surface. Hollow, thin-walled structures were also found. EDS analysis of the elemental content

of area C (Figure 6d) shows that carbon was the most common element on the surface of the sample (70.48 wt%). This means that the observed nano needle-like structures were encapsulated by fixed carbon, which prevented them from being corroded by NaOH. Silicon only comprised 2.30% of the mass fraction, indicating that NaOH had a good removal effect on silica. The sample was also found to contain small amounts of Na, K, and Al. Comparing these results with the elemental composition data presented in Table 1 reveals that Na and K represent impurities remaining in the mineral structure after chemical treatment with NaOH; aluminum originated from the original ore sample. The Al content increased significantly with decreasing Si content. These results show that NaOH was able to remove most of the silicon. Although some impurities remained (in the form of ions), this method still helped reveal the morphology of fixed carbon within the ore.

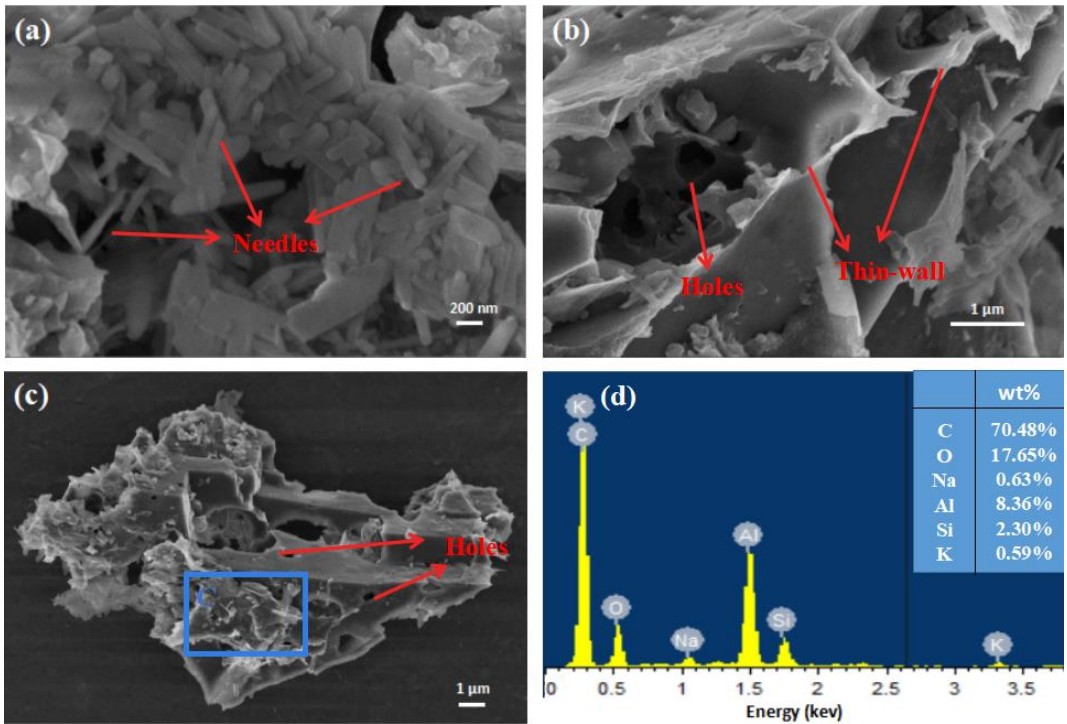

**Figure 6.** (**a**–**c**) SEM images of quartz removal sample and (**d**) EDS elemental spectrum of area C.

### 3.3. Particle Size Distribution

Particle size distribution of the sample is shown in Table 2 and Figure 7. The average particle sizes (D50) of the ore sample, carbon-free sample, and quartz-free sample were 13.80, 6.585, and 11.59 μm, respectively. Particles in the ore sample in the size ranges of 1–45 and <1 μm accounted for 79.92% and 6.48% of all particles, respectively.

**Table 2.** Particle size distributions of ore, carbon-free, and quartz-free samples.

| Particle Size | <1 μm | 1–10 μm | 10–45 μm | >45 μm |
|---|---|---|---|---|
| Ore sample | 6.48% | 34.60% | 44.60% | 13.60% |
| Carbon removal sample | 11.11% | 50.10% | 33.05% | 5.74% |
| Quartz removal sample | 4.00% | 41.02% | 43.76% | 11.22% |

Particles in the carbon removal sample within the size ranges of 1–45, <1, and >45 μm accounted for 83.15%, 11.11%, and 5.74% of the total particles, respectively. Compared with the ore sample particle size distribution (D10, D50, and D90), the particle size of carbon removal sample decreased obviously.

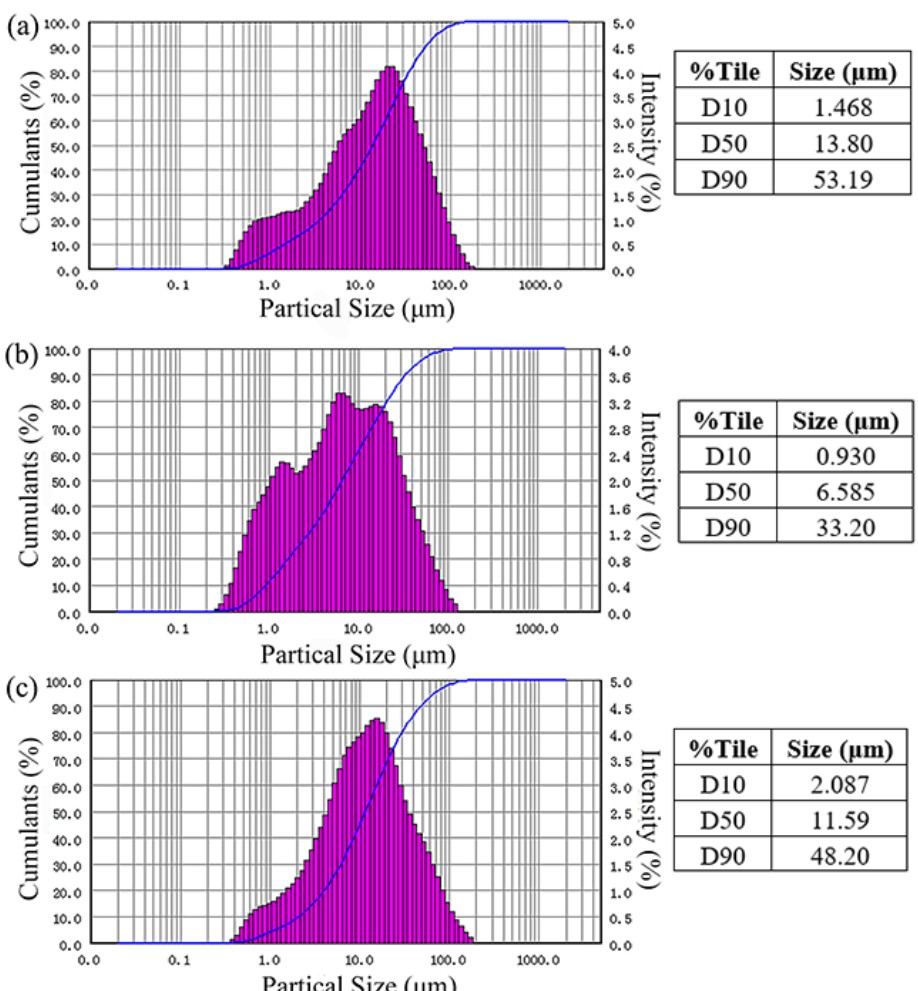

**Figure 7.** Particle size distribution histograms and cumulative particle size distribution (D10, D50, and D90) of (**a**) ore sample, (**b**) carbon-free sample, and (**c**) quartz-free sample.

In the quartz-free sample, the particles were mainly distributed in the range of 1–45 μm (84.78%). Compared with the ore sample, there were fewer particles <1 μm in the quartz-free sample (4%). The decrease in particle size was not obvious.

## 4. Discussion

### 4.1. The Form of Carbon

Combining the DSC-TG analysis results with the multi-element analysis results suggests that the reduced mass (10.79%) was close to the carbon content of the ore (10.77%). There was only one infrared (IR) absorption peak of an organic functional group in the IR spectrum, and only one exothermic peak was observed in the DSC analysis. Thus, the carbon in the ore was most likely present in the form of amorphous carbon rather than as carbonate or organic matter.

### 4.2. Particle Size Distribution Analysis

The particle size distributions were combined with the SEM analysis results, and the reason of the obvious particle size reduction of the carbon removal sample was determined. The carbon-free sample mainly comprised quartz. As the carbon on the quartz surface was removed, the quartz particles were dispersed, because of which, the average particle size of this sample was lower than the average particle size of the ore sample.

The phenomenon of the non-obvious decrease in the particle size of quartz removal sample can also be explained. Because the carbon content of the ore sample increased

following the removal of quartz, the remaining sample flocculated more readily in water. SEM analysis results showed that this sample had lots of holes and that most particles had hollow, thin-walled structures. Thus, the decrease in particle size was not obvious.

### 4.3. Ore's Genesis and Species

According to the above results, the main components of the ore sample were quartz and carbon. Fixed carbon was mainly attached to quartz surfaces. Furthermore, the quartz structure was abnormal, with few edges and corners. This proved that the ore was formed by biological mineralization.

The ore was formed by the accumulation and consolidation of large numbers of phytoliths [21,22]. The ore genesis was inferred as follows: the Shilukeng mining area is located in a warm wetland environment. During the dry season, the lake's water level decreases, turning the area into wetlands where many herbaceous plants grow. During the rainy season, the water level increases, submerging and killing these plants; plant remains thus accumulate year after year. The insoluble silicic acid and stable carbon deposited in these plant cells are preserved and form semi-consolidated phytoliths. These then became sedimentary strata through compaction. The combination of SEM and particle size analysis results reveals that the average particle size of the ore was 13.80 μm, with quartz and carbon existing as nano needle-like structures and nanoparticles. This potential ore was named micro/nano silicon-carbon ore.

### 4.4. Prospective Processing Method

The multi-element analysis indicates that the micro/nano silicon-carbon ore has a high content of quartz and 10.77% carbon. Economically and effectively separating carbon and quartz from the ore is thus essential to extract this ore. However, the carbon and quartz in the ore are closely combined, because of which, quartz and carbon cannot be separated by flotation directly.

Wet grinding is the first step to purify this ore [23]. This method can decrease particle size and separate the carbon from the surface of quartz. Thereafter, flotation is used to separate the quartz and carbon efficiently. To obtain fine-grained high-purity quartz, acid leaching is used to remove the clay minerals. For example, hydrochloric acid, hydrofluoric acid, or nitric acid can dissolve kaolinite, pyrite, and muscovite, for improving the purity of quartz [24]. Thus, acid leaching is an essential step for obtaining fine-grained high-purity quartz. The carbon can be recycled and further purified with sodium hydroxide or hydrofluoric acid.

## 5. Conclusions

The ore was found to mainly comprise α-quartz, fixed carbon, kaolinite, pyrite, and muscovite. Silica accounted for 81.28% of the sample, with carbon, $Al_2O_3$, and $FeS_2$ accounting for 10.77%, 4.56%, and 2.01%, respectively. Ore particle size was mainly distributed between 1.5 and 70 μm, with average particle size of 13.80 μm.

The SEM-EDS results showed that the ore morphology was uneven. Quartz was encapsulated by fixed carbon, and there were obvious holes and depressions on the ore surface. Nano-sized needle-like structures and nanoparticles were also found. These results proved that the ore was formed by biological mineralization.

Based on the findings of this study, this ore may have formed via the deposition of phytoliths in a semi-consolidated state following the death of herbaceous plants in a wetland environment. Insoluble silicic acid and stable carbon were retained in this environment.

According to the morphological characteristics and chemical EA of the ore, the difficulty of purifying the quartz within the ore was found to mainly lie in separating quartz from fixed carbon. The removal of impurities, such as kaolinite, pyrite, and muscovite, was also challenging. Due to its small particle size and high silicon content, this micro/nano silicon-carbon ore has high application potential. In the future, this ore can be used to

obtain fine-grained high-purity quartz, without the need for crushing. The as-obtained quartz has applications in various fields such as electronic information, photovoltaic power generation, and the development of new materials [25–27]. The carbon tailings produced during separation can also be used in rubber reinforcing filler, water purification, or as a conductive material [28,29].

**Author Contributions:** Data curation, H.S., R.Z. and X.W.; funding acquisition, G.D. and X.W.; investigation, S.T. and H.S.; methodology, S.T., H.S., R.Z. and J.W.; resources, S.T. and J.W.; writing–original draft, S.T.; writing–review and editing, G.D. All authors have read and agreed to the published version of the manuscript.

**Funding:** This research and the APC was funded by Jiangxi Geological Exploration Fund Project grant number 20220015.

**Data Availability Statement:** Not applicable.

**Conflicts of Interest:** The authors declare no conflict of interest.

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
