# Peer review of "Process Mineralogy of Micro/Nano Silicon-Carbon Ore Obtained from Jiangxi, China"

_minerals, doi:10.3390/min12060700_

Round 1
Reviewer 1 Report
Dear authors, I have added all my comments on the manuscript as well as on the reviewer report.

Author Response
We would like to thank you for your careful reading, helpful comments, and constructive suggestions, which has significantly improved the presentation of our manuscript.
We have carefully considered all comments from the reviewers and revised our manuscript accordingly. The manuscript has also been double-checked, and the typos and grammar errors we found have been corrected. In the following section, we summarize our responses to each comment from the reviewers. We believe that our responses have well addressed all concerns from the reviewers. We hope our revised manuscript can be accepted for publication.
We submitted the cover letter in the attachment.

Reviewer 2 Report
In this manuscript, the authors used a variety of analytical test methods to analyze the unknown ore’s genesis, species and application field. These results help to provide a theoretical basis for the investigation, purification, and utilization of the ore in question. In general, it has certain practicability and innovation, but there are still some problems to be solved in order to meet the publication requirements of the journal :
1. The writing and presentation should be significantly improved. For example, I don’t think it's appropriate to use too many complex sentences. The authors are advised to carefully go over the whole manuscript and to rephrase any sentences or terms wherever needed for accurate and better presentation.
2. The authors should pay more attention to details. For example, The 1 in line 52 of the second page should be changed to a 2, which is the title of the second section Materials and methods; A space should be added between the 20mg data and the unit in the 88th line of the second page. The same problem is the 95th line of the third page. Please check whether the unit format of the full text is correct?
3. Data 796.46 and 777.17 in line 154 on page 4 do not match data 796.16 and 771.17 in Figure 2b, check for the same problem elsewhere?
4. The closer the particle size distribution map is to the normal distribution, the more uniform the particle size distribution is. None of the three samples in Figure 6 showed a good normal distribution. How about the sample distribution uniformity and applicability? Please provide D10, D50, D90 data for judging sample quality.
My general opinion is receiving after minor revisions.

Author Response

(The authors gave the same response as above.)

Reviewer 3 Report
Dear Authors,
I suggest to improve methodology, there are not to much infrormation of samples in differetent type of methods.
page 3 line 95: which kind of FTIR measurements were done? Please ad more details to methodology.
page 3 methodology: How much samples were analyzed in each methods?
Page 5 figure 2: regarding to shape of spectrum, the y axis seems rather absorbance, not transmittance.
see an example spectrum here:
Author Response

(The authors gave the same response as above.)

Round 2
Reviewer 1 Report
Dear authors,
The revised manuscript has improved significantly, but certain aspects that I addressed in the first review were not addressed. I have indicated these in the annotated file. The results section still contains interpretations which should be in the discussion section. There also needs to be a link between the results and the ore genesis which is mentioned in the discussion, for example how do the results support your ore genesis discussion? The ore genesis discussion is only supported by the SEM results, what about the FTIR results, ICP-MS etc.

Author Response
We would like to thank you for your careful reading, helpful comments, and constructive suggestions, which has significantly improved the presentation of our manuscript.
We have carefully considered all comments from the reviewers and revised our manuscript accordingly. The manuscript has also been double-checked, and the typos and grammar errors we found have been corrected. In the attachment, we summarize our responses to each comment from the reviewers. We believe that our responses have well addressed all concerns from the reviewers.We hope our revised manuscript can be accepted for publication.
Please see the attachment and manuscript.
